# Empowering Health Workers to Protect their Own Health: A Study of Enabling Factors and Barriers to Implementing HealthWISE in Mozambique, South Africa, and Zimbabwe

**DOI:** 10.3390/ijerph17124519

**Published:** 2020-06-23

**Authors:** Elizabeth S. Wilcox, Ida Tsitsi Chimedza, Simphiwe Mabhele, Paulo Romao, Jerry M. Spiegel, Muzimkhulu Zungu, Annalee Yassi

**Affiliations:** 1School of Population and Public Health, University of British Columbia, Vancouver, BC V6T 1Z3, Canada; jspiegel@mail.ubc.ca; 2International Labour Organization, Harare, Zimbabwe; chimedza@ilo.org; 3International Labour Organization, Decent Work Team for East and Southern Africa, Pretoria 0020, South Africa; mabhele@ilo.org; 4International Labour Organization, Maputo, Mozambique; romao@ilo.org; 5National Institute for Occupational Health, a division of the National Health Laboratory Service, Johannesburg 2001, South Africa; MuzimkhuluZ@nioh.ac.za; 6School of Health Systems and Public Health, University of Pretoria, Pretoria 0001, South Africa

**Keywords:** occupational health, i-PARiHS framework, health workers, HealthWISE, implementation science

## Abstract

Ways to address the increasing global health workforce shortage include improving the occupational health and safety of health workers, particularly those in high-risk, low-resource settings. The World Health Organization and International Labour Organization designed HealthWISE, a quality improvement tool to help health workers identify workplace hazards to find and apply low-cost solutions. However, its implementation had never been systematically evaluated. We, therefore, studied the implementation of HealthWISE in seven hospitals in three countries: Mozambique, South Africa, and Zimbabwe. Through a multiple-case study and thematic analysis of data collected primarily from focus group discussions and questionnaires, we examined the enabling factors and barriers to the implementation of HealthWISE by applying the integrated Promoting Action on Research Implementation in Health Services (i-PARiHS) framework. Enabling factors included the willingness of workers to engage in the implementation, diverse teams that championed the process, and supportive senior leadership. Barriers included lack of clarity about how to use HealthWISE, insufficient funds, stretched human resources, older buildings, and lack of incident reporting infrastructure. Overall, successful implementation of HealthWISE required dedicated local team members who helped facilitate the process by adapting HealthWISE to the workers’ occupational health and safety (OHS) knowledge and skill levels and the cultures and needs of their hospitals, cutting across all constructs of the i-PARiHS framework.

## 1. Introduction

Health workers (HWs) are in short supply worldwide. It is estimated that by 2030, the global health workforce will be short approximately 18 million workers, primarily in low- and middle-income countries (LMICs) [1]. In high-risk settings, where disease prevalence is high and health systems are stretched to provide basic health services, HWs are at an elevated risk of contracting infectious diseases such as hepatitis, human immunodeficiency virus (HIV), tuberculosis (TB), and novel emerging threats from occupational exposure, including COVID-19. They also suffer stigma and discrimination at work, in their communities, and at home from bearing these increased risks. 

Effective coverage by the health workforce depends on availability, accessibility, acceptability, and quality of HWs [2]. One of the top three factors reducing supply, along with migration and retirement, is the “risk of violence, illness or death” [3]. Strategies to address the HW shortage ought to therefore include protecting HWs by promoting their health and safety at work, particularly in high-risk settings. 

International organizations have developed tools to improve the occupational health and safety (OHS) of HWs. One of these is HealthWISE, a participatory, quality improvement tool, jointly developed by the International Labour Organization (ILO) and the World Health Organization (WHO) [4]. In 2010, a tripartite group consisting of workers’, employers’, and governments’ representatives, as well as specialists from the ILO and WHO, convened and agreed on a framework for improving the OHS of HWs. Based on principles from the original Work Improvement in Small Enterprises (WISE) training program created by the ILO, HealthWISE was then developed to help support the implementation of this framework [4]. HealthWISE aims to improve working conditions, performance, and workplace safety through training and empowering HWs with the ability to identify workplace hazards and areas requiring improvement in their work environments and to conduct processes for developing and implementing low-cost solutions to address them. HealthWISE consists of two workbooks, one for participants and one for trainers, with content organized into eight modules (Figure 1). The workbooks are available online in five languages. As with addressing the supply of HWs, the availability of the tool is only part of the solution. It is also important to understand the implementation of HealthWISE and to improve upon implementation processes to maximize the tool’s potential.

Implementation science is growing in the field of global health. Madon and colleagues [5] called on researchers to (i) “develop theoretical models and new analytic methods that apply to resource poor settings” such as areas where HWs are in short supply, (ii) build capacity and strengthen research institutions in LMICs in regard to implementation science, in part to learn from valuable local knowledge and insights that influence implementation processes, and (iii) increase collaboration with governments, non-governmental organizations, and communities to incorporate research into implementation processes in order to improve upon them. 

This paper overviews the implementation of three HealthWISE modules in seven hospitals (designated the letters A through G) in Mozambique, South Africa, and Zimbabwe. Using the integrated Promoting Action on Research Implementation in Health Services (i-PARiHS) framework (described in “Research Methods” below), it aims to better understand the enabling factors and barriers to the implementation of HealthWISE in these hospitals and, considering previous implementation science research, to discuss how these might be leveraged or overcome in future implementations of HealthWISE.

## 2. Materials and Methods 

### 2.1. Country Contexts

An existing North South partnership involving researchers and technical teams from Canada and South Africa [6] was expanded to include team members from Mozambique and Zimbabwe due to plans and interest to implement HealthWISE in those countries. This enabled a comparison of its implementation in different contexts. The three countries are in close proximity in the southern African region and represent high-risk settings where the OHS of HWs is at different stages and resource levels and has yet to be fully given the importance that it is due. According to 2018 World Bank classifications based on gross national income (GNI) per capita, Mozambique is a low-income country (GNI per capita of US$460), South Africa an upper-middle-income country (GNI per capita of US$5750), and Zimbabwe a lower-middle-income country (GNI per capita of US$1790) [7]. Total health expenditures per capita (and as a percentage of gross domestic product) from 2017 reflect this trend, with Mozambique spending US$21.07 (4.94%), South Africa US$499.24 (8.11%), and Zimbabwe US$110.15 (6.64%) [8]. Mozambique, South Africa, and Zimbabwe are amongst 30 high burden countries with regard to TB, TB and HIV co-infection, and multi-drug resistant TB [9]. In 2018, the incidence of TB per 100,000 people was 551 in Mozambique, 520 in South Africa, and 210 in Zimbabwe. The total prevalence of HIV among their populations aged 15-49 was 12.6% in Mozambique, 20.4% in South Africa, and 12.7% in Zimbabwe [8]. 

### 2.2. Implementation of HealthWISE

A planning meeting with representation from all countries was held in Zimbabwe in February 2016. At this meeting, it was decided that the focus would be restricted to specific, related priority areas: biological hazards and infection control (Module 3) and discrimination, harassment, and violence (Module 4). Over the following eight months, local team leads sought any necessary local, provincial, and national approvals and selected hospitals in which to implement HealthWISE. A total of seven hospitals participated: three in Mozambique, two in South Africa, and two in Zimbabwe. Implementation refers to the ensuing activities, including the introduction of HealthWISE, by training groups of HWs at participating hospitals and the activities carried out by participants from this point through to the final capstone meeting. Observation focused on if and how participants used HealthWISE in their hospitals and included the activities conducted by research team members, such as focus groups and questionnaires, to inquire into the enabling factors and barriers to its uptake and resultant activities.

Implementation was observed over 20 months, beginning with three Training-of-Trainers (ToT) workshops (one per country) in October and November 2016. The three-day program was developed and carried out by local team members. In brief, the focus of day 1 was on introducing HealthWISE and Modules 1 and 3. The focus of day 2 was on Module 4. Day 3 was devoted to developing HealthWISE action plans (activities to be carried out by trainees in their health facilities). Throughout, participatory training techniques, including role plays and an interactive exercise on the topic of stigma [10], were demonstrated, which might be useful for participants to help engage workers and disseminate new information in their workplaces. Following the training, participants were expected to create HealthWISE teams and based on the HealthWISE principle of finding simple, low-cost solutions to workplace issues within their local contexts, carry out HealthWISE activities in their hospitals. Based on budgets determined by the action plans, a small amount of project funds was made available for HealthWISE activities. Additional practical training sessions were held in Mozambique in July 2017 and in Zimbabwe in February 2018. The HealthWISE teams trained during the ToT workshops conducted walk-through assessments with the lay HWs in one or more departments, helping to identify hazards and how they might be mitigated with low- or no-cost solutions.

One year after the ToT workshops, six follow-up workshops were held (one per hospital, with hospitals A and C in Mozambique combined). During these workshops, participants presented on HealthWISE activities that had taken place in their facilities to date and participated in focus groups on the perceived enabling factors and barriers to implementing HealthWISE. Participants were asked to individually brainstorm their own lists of enabling factors and barriers and to then read these out one-by-one and explain them to the group. Questions and discussion were encouraged. Participants in Mozambique and South Africa also completed an anonymous questionnaire. Due to resource constraints within the hospitals, these questionnaires were unable to be administered in Zimbabwe. Shortly after the follow-up workshops, one representative from each hospital from all three countries attended a dissemination meeting in South Africa to present on their progress implementing HealthWISE.

The research focus of the project culminated in a final meeting in May 2018 in Zimbabwe (Figure 2). One representative from each hospital presented on HealthWISE activities that had taken place to date and provided feedback on the implementation process and preliminary findings from the research.

### 2.3. Research Methods

A multiple-case study, in which each hospital was treated as a single case, was used to examine the enabling factors and barriers to the implementation of HealthWISE [11]. Within the case study, thematic analysis, “a method for identifying, analysing and reporting patterns (themes) within data”, was used [12].

The integrated Promoting Action on Research Implementation in Health Services (i-PARiHS) framework was employed in this study [13]. The i-PARiHS framework was published in 2016, based on an earlier iteration from 1998 [14] and continues to be developed and refined. The framework describes four constructs related to implementation: (i) the ‘innovation’, or new knowledge informed by evidence-based research, that is being introduced; (ii) the ‘recipients’, or the individuals and teams who are involved in or affected by the implementation; (iii) the ‘context’, referring to three levels of local, organizational, and external health system settings in which the innovation is being implemented; and (iv) ‘facilitation’, or the strategies and actions performed by the facilitator(s) to enable implementation in response to the innovation and its recipients within their given context. The earlier version was classified as an explanatory framework that specified the relationship between the constructs [15] and while the integrated version maintains these linkages, the i-PARiHS framework is also descriptive as it breaks-down the constructs to further describe characteristics important to implementation. Descriptive and explanatory frameworks are used to understand factors that might have positively or negatively influenced implementation processes [16] and given its continuing evolution, the i-PARiHS framework was chosen to explore the implementation of HealthWISE. It ought to be noted that several tools have been developed based on the original PARiHS framework to more thoroughly assess the context construct, including the Context Assessment for Community Health (COACH) tool, specifically for use in LMICs [17]. While the dimensions described therein were considered during this analysis, they are captured in the more recent i-PARiHS framework and the characteristics of the latter were therefore used.

Data were drawn from dissemination and capstone meeting presentations (PowerPoint presentations), focus group transcripts (Word documents generated from audio-recordings), and open-ended responses to completed anonymous questionnaires (paper and electronic PDF copies). Using an inductive approach, three focus group transcripts, one from each country, were first open coded to generate a list of enabling factors and barriers. These codes were then compared and categorized according to the i-PARiHS constructs and characteristics to generate a draft codebook. The remaining three focus group transcripts were then coded using the draft codebook. Some characteristics were subsequently removed or combined to refine the constructs and characteristics to those listed in Tables 2–5, which are the themes and sub-themes for the final codebook. Using this codebook, all data were coded using NVivo 12 and Excel to better understand the enabling factors and barriers to the implementation of HealthWISE. 

This study was approved by the Behavioural Research Ethics Board, University of British Columbia, Canada (H17-00286, H17-00039), the Research Ethics Committee, University of Pretoria, South Africa (159/2017), and the Medical Research Council of Zimbabwe, Zimbabwe (MRCZ/A/2240). Participants who were involved in the follow-up workshop focus groups, questionnaires, and dissemination and capstone meetings were provided with written information about the research objectives and processes prior to their involvement and individual informed consent was obtained. Participation was voluntary and individuals were informed of their right to withdraw from the study at any time. All data was collected anonymously or de-identified before analysis to protect confidentiality. 

## 3. Results

### 3.1. Hospital Characteristics

The seven participating hospitals ranged in size from 36 to 1652 beds and from approximately 139 to 4407 workers. In all hospitals, the workers were predominantly female. Characteristics of the seven hospitals are presented in Table 1.

### 3.2. Enabling Factors and Barriers

Results for the four constructs—innovation, recipients, context, and facilitation—are presented below, with quotes that help to reflect what was an enabling factor or barrier in the implementation of HealthWISE in each of the participating hospitals in Mozambique, South Africa, and Zimbabwe. 

It is important to note that not all constructs or characteristics were explicitly mentioned by participants at each of the hospitals. This absence does not necessarily mean that a specific characteristic was or was not an enabling factor or barrier; while this could be the case, it could instead indicate that further questioning on specific characteristics of interest may be warranted in future studies. Where a characteristic is designated as both an enabling factor and barrier (EF/B) within the tables, further information on how the implementation of HealthWISE was helped or hindered is provided in the ensuing description.

#### 3.2.1. Innovation

The innovation construct included characteristics related to HealthWISE, the intervention being implemented. Three of the seven characteristics of the innovation construct from the i-PARiHS framework were mentioned; whether they were enabling factors and/or barriers in each of the hospitals is shown in Table 2.

“Clarity” about HealthWISE—what the tool was and why and how it was going to be used—was a key enabling factor in nearly all of the hospitals: *"When personnel have been trained and they know… what is expected of them and what is going to be done, they are more cooperative than when they do not know"* (Hospital E, Focus group). The “relative advantage” of HealthWISE—how it would be of benefit compared with existing interventions—particularly that the tool aimed to benefit workers and their working environment (as opposed to being focused solely on patients), helped to spur the implementation of HealthWISE in one hospital in Mozambique and one hospital in South Africa where the *“anticipated positive results/effects of [the] HealthWISE project”* (Hospital D; Capstone meeting) were an enabling factor. “Observable results” were also mentioned as enabling the implementation of HealthWISE by one hospital in each country.

In opposition, one barrier of the innovation construct, mentioned by all the hospitals in Mozambique, was a lack of clarity or *“lack of knowledge about exactly what to do”* (Hospital A, Questionnaire). For some hospitals, there was also an inability to raise awareness about HealthWISE among hospital staff: *“...we did not have much time to publicize this project to colleagues to understand what it was all about”* (Hospital B, Focus group). One hospital in South Africa echoed this lack of clarity and awareness, as represented by one comment of *“supervisors not understanding the project”* (Hospital E, Questionnaire). 

#### 3.2.2. Recipients

The recipients construct included characteristics related to the individuals and teams involved in, or affected by, the implementation of HealthWISE. The 11 characteristics of the recipients construct and whether they were enabling factors and/ or barriers in each of the hospitals are shown in Table 3. The “time, resources, support” characteristic from the i-PARiHS framework was split into four: “project funding”, “human resources”, “material resources”, and “personal protective equipment (PPE)” to better capture their different impacts on the implementation of HealthWISE. Two characteristics from the i-PARiHS framework, “values and beliefs” and “presence of boundaries”, did not emerge during this analysis.

“Motivation”, or the *“willingness of staff to participate in HealthWISE activities”* (Hospital C, Capstone meeting), was mentioned as an enabling factor by hospitals C and F, where workers *“showed much interest in this program”* (Hospital F, Focus group). A lack of external incentives was mentioned as a barrier by both Zimbabwe hospitals F and G, where *“there is in most cases lack of incentives for trainers to keep their motivation high”* (Hospital F, Dissemination meeting).

Lack of clear “goals and expectations” related to the implementation of HealthWISE was mentioned as a barrier by the three Mozambique hospitals A, B, and C, where there was *“difficulty of perception of some professionals about the objectives of the project”* (Hospital B, Dissemination and Capstone meetings) as well as *“failure to comply with agreed deadlines”* (Hospital A, Dissemination meeting). On the other hand, having clear “goals and expectations” was an enabling factor for both hospitals D and E in South Africa and for hospital G in Zimbabwe. Goals came in different forms, such as an *“attainable objective that was set by the team member”* (Hospital D, Focus group), as well as *“an action plan that served as our guiding point of reference”* (Hospital E, Focus group).

Individual “skills and knowledge” related to OHS was both an enabling factor and barrier mentioned by hospitals A, B, and C in Mozambique, depending on whether recipients were perceived as having or lacking OHS knowledge. There was some overlap with the culture characteristic in the context construct, which was discussed to a greater extent in Zimbabwe and South Africa.

Lack of “project funding” was mentioned as a barrier by all but one hospital in Mozambique. *“Lack of resources for implementation”* (Hospital A, Dissemination meeting), *“financial constraint”* (Hospital E, Questionnaire), and *“lack of funding for full implementation of HealthWISE”* (Hospital F, Capstone meeting) were some of the ways that this barrier was mentioned. Since this initially proved to be a major constraint, it was addressed by local team members in all three countries through communication and practical training sessions that helped to redirect workers towards no-cost solutions. For example, a patient consultation room was rearranged to improve ventilation and reduce the risk of workers being exposed to airborne pathogens and, in the same area, ripped flooring that created a fall hazard was cut out and removed, as opposed to being replaced with new flooring (Figure 3). Workers involved in the practical training sessions expressed that *“the search for solving problems that do not require financing was a great gain”* (Hospital A, Focus group). Where funding for larger project activities was available, it was an enabling factor. Hospital C in Mozambique was able to draw from external funding sources to begin construction of a new TB consultation and testing unit and laboratory when the old infrastructure was identified as a hazard by the HealthWISE team. One hospital in Zimbabwe, which used project funds to purchase some equipment for their training sessions, indicated that the *“allocation of funds”* (Hospital F, Dissemination meeting) was an enabling factor to implementation in their hospital.

“Human resources” were a barrier mentioned by all hospitals except A and C in Mozambique. All hospitals seemed to experience some degree of staff shortages, *“when the departments are so short-staffed, they are reluctant to take part in some of the activities and to attend some of the meetings”* (Hospital E, Focus group); turnover, *“staff movement, some people are exiting the system, others might be on night duty, you know, on leave”* (Hospital D, Focus group); overwhelming *“workload from the department”* (Hospital D, Focus group); and lack of time, *“we always have quite demanding tasks that we do every day, our jobs are quite demanding, so the lack of time maybe is one of the major barriers to the implementation”* (Hospital F, Focus group). 

“Material resources”, including the availability of reference, training, and other materials for the practical application of HealthWISE were mentioned by all participating hospitals. Where material resources were available, they were an enabling factor; where they were unavailable, they were a barrier. There was also an issue with the *“scarcity of surgical medical material with an emphasis on personal protective equipment”* (Hospital B, Capstone meeting) mentioned by the three hospitals in Mozambique, however there were indications from one hospital in each South Africa and Zimbabwe that the process of implementing HealthWISE was helping to secure *“some improvements in procurement e.g., availability of appropriate PPE for linen bank staff and food services personnel and respirators”* (Hospital E, Capstone meeting) and that *“if we procure that [PPE] then it will be, the HealthWISE program will be effective”* (Hospital G, Focus group).

Where active and engaged, the “local opinion leaders”, referring to the existing OHS teams or newly created HealthWISE teams, were an enabling factor. This was particularly the case at both hospitals in South Africa, where OHS teams were in place before the implementation of the project. One was commended as a *“knowledgeable, skilled, reliable and committed OHS team”* (Hospital D, Capstone meeting) and the other as a diverse *“HealthWISE team comprising of members from different departments e.g., HR [human resources], staff development, cleaning, IPC [infection prevention and control] and linen bank”* (Hospital E, Capstone meeting), which helped contribute to their success. The smaller hospital in Zimbabwe also noted their *“dedicated HealthWISE champions”* (Hospital G, Capstone meeting). The barrier was due to *“peripheral involvement of medical doctors and the nurses’ representative member”* (Hospital F, Capstone meeting), again indicating that engagement of diverse teams was one of the keys to successful implementation.

“Collaboration and teamwork” were mentioned as enabling factors or, where lacking, as barriers by the three hospitals in Mozambique. Similar to the motivation characteristic or the willingness of the workers to engage in the HealthWISE project, this characteristic referred to the involvement and inclusion of workers. Hospitals A and C felt that there was both *“good participation and adherence from employees”* as well as that *“there must be greater involvement of employees in the HealthWISE project”* (Hospital A/C, Focus group). 

“Existing networks”, referring to collaboration and communication within and between hospitals, was generally an enabling factor mentioned by six of the seven hospitals. Communication, expressed as *“the exchange of information among workers and from workers to patients; Reciprocal information sharing”* (Hospital A/C, Questionnaire) was key, as was the *“easy implementation and dissemination of information to colleagues”* (Hospital B, Dissemination and Capstone meetings). One participant from South Africa mentioned that *“HR has been absolutely amazing. Having a member of HR in our HealthWISE team was the best thing”* (Hospital E, Focus group), due to the improved communication between departments and with management that enabled more project activities to receive approval and take place. Finally, the *“collaboration between [Hospital A] and [Hospital C] - exchanged experiences and helped to overcome difficulties that were encountered”* (Hospital A/C, Dissemination meeting) was particularly helpful. The barrier in this regard was *“poor communication”* (Hospital D, Capstone meeting).

“Power and authority” were identified as enabling factors for hospitals A, C, and G. This generally related to workers feeling empowered to take charge of their own health and safety. During the focus group, one participant mentioned the idea of greater ownership over their own safety: *“For me the project came to change my way of thinking... I realized that I am able to improve… my safety in the workplace and not wait for the bosses to come to control something within the sector, and so it was positive for me”* (Hospitals A/C, Focus group). This sentiment was shared by a participant in Zimbabwe: *“...when we started it was your program but now slowly it is becoming our program so if everyone is involved at that level then we are going to succeed”* (Hospital G, Focus group). These feelings also manifested as achievements; at Hospital C in Mozambique, one team member used material from Module 5, which was not part of the initial training, and worked and negotiated with management and the local municipality to more routinely dispose of waste that piled up on the hospital grounds. A *“lack of authority to implement certain activities”* (Hospital E, Capstone meeting) was a barrier mentioned by one hospital in South Africa. 

#### 3.2.3. Context

The context construct included characteristics related to the setting in which the innovation was to be implemented. Characteristics related to the local and organizational levels from the i-PARiHS framework were combined as these were difficult to piece apart and included six characteristics. The external health system level included three of five characteristics from the i-PARiHS framework, leaving out “policy drivers and priorities” and “incentives and mandates”. Identified enabling factors and/or barriers in each of the hospitals are shown in Table 4.

“Senior leadership and management support” was an enabling factor mentioned by all seven hospitals through comments such as *“support from the management... and the participation of those in charge of the sectors”* (Hospital B, Focus group), *“buy-in from senior management”* (Hospital D, Focus group), and *“management acceptance of the program”* (Hospital G, Focus group). *“Lack of support from some middle managers”* (Hospital D, Capstone meeting) where, for instance, heads of departments were at times not willing or able to release workers from their duties to participate in HealthWISE activities, was a barrier in hospitals D, E, and F.

Elements of the “culture” characteristic were mentioned by five of the participating hospitals. Where workers were perceived to have greater “commitment to work”, with descriptions such as *“strong workforce”* (Hospital G, Focus group), this was designated as an enabling factor. In Mozambique, *“one of the barriers [was] that information [had] to be oral”* since workers were *“not in the habit of stopping to read”* (Hospital B, Focus group). Here, there were overlaps with the willingness of workers to participate and learn and the degree of teamwork and collaboration described in the recipients construct above. Where difficulties were raised in regard to “knowledge application”, this was designated as a barrier. In the two hospitals in South Africa, a *“lack of safety culture; lack of knowledge about the importance of OHS matters and the HealthWISE program”* (Hospital E, Capstone meeting) was discussed as a barrier, referring to the idea that workers have OHS *“knowledge but they are not interested [in applying it]”* (Hospital E, Focus group). *“Negative hospital staff attitudes”* (Hospital D, Capstone meeting) and *“resistance to change”* (Hospital B, Questionnaire) were also perceived as a barrier in several hospitals. One hospital in Zimbabwe that had expanded the services of their regular staff wellness clinic as part of the implementation of HealthWISE also mentioned *“fear to uptake services… due to fear of stigma and discrimination”* (Hospital G, Capstone meeting). 

Competing “organizational priorities” and programs were mentioned as a barrier by hospitals D, F, and G. Hospital A seemed to have priorities and programs that served to support, instead of compete with, HealthWISE activities, indicating *“reinforcement of the ongoing IPC activities; Synergies have been built among HealthWISE and IPC”* (Hospital A, Dissemination meeting).

In regard to “structure and systems”, infrastructure was a barrier mentioned by all hospitals. Infrastructure was generally older and difficult to change and participants seemed to feel that *“some infrastructure hinders the proper functioning of the project”* (Hospital A/C, Focus group), such as the *“lack of ramps to move trolleys”* (Hospital B, Dissemination and Capstone meetings) at one hospital in Mozambique and that *“buildings were not constructed in such a way that they allow for proper ventilation”* (Hospital F, Focus group) at one hospital in Zimbabwe. Hospital G expressed that the *“infrastructure… might not be ideal but we are going to work with what we have”* (Hospital G, Focus group). *“Lack of a specific project space”* (Hospital A/C, Focus group) was mentioned by one hospital in Mozambique, while in South Africa, both hospitals perceived the lack of a dedicated OHS clinic *“where you can see your employees when they are sick”* (Hospital D, Focus group) or *“lack of equipment at our OHS clinics”* (Hospital E, Questionnaire) as barriers as well. Institution size, specifically in relation to the number of workers trained on HealthWISE, was also a barrier specifically mentioned by Hospital F in Zimbabwe.

Regarding “history of innovation and change”, resistance to change was mentioned as a barrier by participants at hospitals A, B, C, E, and G. One participant from South Africa indicated that *“anytime there is a new project there will always be resistance… because people are used to the norm of how they usually do things"* (Hospital E, Focus group) and another from Mozambique indicated that *“resistance to change on the part of some colleagues was a challenge in the past”* (Hospital B, Questionnaire). 

“Evaluation and feedback processes” were perceived to be both enabling factors and barriers to the implementation of HealthWISE. Improvements to reporting procedures were an enabling factor in hospital A, demonstrated by the increased *“willingness of staff to communicate accidents in the workplace”* (Hospital A, Capstone meeting), however *“lack of understanding and knowledge of procedures, e.g., incident reporting procedure”* (Hospital D, Capstone meeting) and *“poor reporting of incidents”* (Hospital E, Capstone meeting) remained barriers elsewhere. Lack of feedback was a barrier mentioned by hospitals B, D, and E. Hospital B indicated the *“need to find a way to get worker feedback”* (Hospital B, Focus group) and Hospital D felt that *“if you don’t give feedback to the unit… it may compromise participation at the later stage”* (Hospital D, Focus group).

In regard to external context, existing OHS “regulatory frameworks” were mentioned as an enabling factor by the two hospitals in South Africa, such as the existence of an *“Occupational Health and Safety Act that we need to adhere to”* (Hospital D, Focus group). The “instability of the health system environment” was mentioned as a barrier by Hospital F in Zimbabwe, where significant political and socioeconomic changes and challenges occurred during the course of the project. Having good “inter-organizational networks/relationships”, particularly with trade unions that play more prominent roles in South Africa, was mentioned as both an enabling factor and, where lacking, a barrier by Hospitals D and E. 

#### 3.2.4. Facilitation

The facilitation construct included characteristics related to the strategies and actions performed by the facilitator(s) to enable implementation by adapting HealthWISE in response to the workers who were asked to use it within the contexts of their hospitals and countries. Two characteristics and whether they acted as enabling factors and/or barriers in each of the hospitals is shown in Table 5.

At least one hospital in each country declared that the “HealthWISE trainings” were key enabling factors to the implementation of HealthWISE—that the tool would not have been implemented solely based on the workbooks being available. In the hospitals in Mozambique, an identified barrier was that the *“training time was too short”* (Hospital A/C, Focus group) and was therefore insufficient to support the implementation process. The practical training sessions in Mozambique and Zimbabwe also helped participants with relatively less OHS experience better identify workplace hazards and solutions: *“...what helped was a second meeting [implementation training]... when we realized what in theory we had to do in practice”* (Hospital A/C, Focus group). 

Ongoing “communication and support from the research team” was an enabling factor in five of the participating hospitals. The research team was available, however participants at the different hospitals reached out to varying degrees. In the hospitals in South Africa, one member of the research team was more accessible to participants and was therefore able to more quickly answer questions and on occasion, help to troubleshoot issues that arose. At the same time, key participants at hospitals A and F in Mozambique and Zimbabwe, respectively, were also more engaged and would reach out more often if queries or problems arose and would, in turn, receive desired engagement. For instance, one worker expressed that *“we had a permanent contact with the team that trained us… I would email him, and he would respond quickly”* (Hospital A, Focus group). The research team was therefore accessible, however was used in different ways and amounts by participants at each hospital. Where *“continued interaction between the hospital and the research team”* (Hospital F, Dissemination meeting) took place, it was perceived as an enabling factor. 

## 4. Discussion

Many programs and tools available for health facilities focus on measuring and improving patient health, quality of service, and safety culture and have been reviewed in several publications [18,19]. The few that have been developed and studied related to HWs include online infection control tools [20] and a seasonal influenza vaccination rate improvement tool [21]. HealthWISE is a widely available quality improvement tool that addresses a variety of OHS concerns encountered in health facilities, containing information and activities that are particularly suitable to areas with few resources and little OHS experience. While it has been piloted in several countries, including in Senegal, the United Republic of Tanzania, and Thailand, and has since been implemented in the United States [22], China [23], and the Gambia, few publications and reports discussing or evaluating these experiences are publicly available. This paper is therefore one of the first to detail the implementation of HealthWISE and provide an analysis of enabling factors and barriers encountered in different LMICs during the process. Applying the i-PARiHS framework enabled an identification of key enabling factors as characteristics of the “recipients” and “context” constructs and included the willingness of workers to engage in implementation, the presence of diverse teams that championed the implementation process, and supportive senior leadership. Barriers were reported in all constructs and included a lack of clarity about how to use HealthWISE, insufficient funds, stretched human resources, older buildings, and lack of incident reporting infrastructure. Overall, successful implementation of HealthWISE called for dedicated local research and technical team members who helped facilitate the process by adapting HealthWISE to the workers’ OHS knowledge and skill levels and the cultures and needs of their hospitals, cutting across all constructs of the i-PARiHS framework. 

HealthWISE was well-received in all participating hospitals, demonstrating the importance of the innovation itself. Workers in Hospital A seemed especially interested in a tool that was aimed at their own needs as opposed to solely those of patients. In the majority of hospitals, even those that did not mention it explicitly, there was difficulty in fully understanding HealthWISE and how to use it, as well as how to spread awareness regarding its implementation throughout the hospital. The anticipated benefits or the results that were being observed over the course of the implementation spurred several hospitals forward. The research team felt there was room to improve the messaging related to HealthWISE, and future implementations might further explore how information related to the innovation impacts the implementation process. Questions could explore what factors made HealthWISE more acceptable, such as its development by the ILO and WHO, the quality or contents of the materials, and its fit within existing hospital practices and values in order to explore additional aspects of the innovation construct. 

HealthWISE activities were accomplished in all hospitals due to the active efforts of HealthWISE recipients, the individuals and teams who were overall key to the implementation. Barriers described in this study, including staff shortages, high workloads, and limited material resources, have been identified among common barriers to evidence implementation across clinical areas in LMICs [24]. Despite these issues, workers possessed a wealth of knowledge related to their hospitals and colleagues and therefore knew best when it came to implementing HealthWISE. While participants at hospitals in Mozambique and Zimbabwe mentioned feeling empowered and taking ownership over HealthWISE, more might have been done to help all teams recognize earlier on that their local insights and initiatives were what would make HealthWISE more successful. Literature on team innovation and implementation points to the need for varied team composition to promote creativity and action [25]. Where teams were more diverse, particularly where members helped link frontline workers to management, more activities seemed to receive approval and were able to move forward. Having established OHS teams in both hospitals in South Africa, along with the accompanying knowledge and skills, also enabled them to accomplish a variety of tasks. It seemed that HealthWISE was a catalyst to the implementation of available OHS policies and allowed workers to take ownership of their own health and safety, with the OHS professionals providing oversight. The importance of opinion leaders—“people who influence the opinions, attitudes, beliefs, motivations, and behaviors of others” [26]—has been shown in other examples of successful implementation processes and was similar here [27]. Promoting collaboration and exchange between sites that are implementing the same innovation might be interesting to explore, since this was an enabling factor for Hospitals A and C who worked together in Mozambique and was of particular interest to participants in Zimbabwe who expressed a desire to visit other sites and learn from them during the implementation process. 

Implementing HealthWISE in hospitals in different countries enabled the exploration of several characteristics of the context construct, which in these low-resource settings were more often barriers to the implementation of HealthWISE rather than enabling factors. Context has repeatedly been discussed as an important factor in regard to implementing various interventions, one that deserves more thorough definition and analysis [28]. Findings from this study may help to better identify which characteristics are most important to the implementation of HealthWISE and, to some degree, other similar projects in LMICs. In terms of the context construct, local and organizational factors were more often discussed than external health system ones. 

While senior management were supportive of the project in all hospitals and welcomed its implementation, middle management were perceived as a barrier at several sites. Engaging more actively with middle management to explore strategies that would have allowed workers to be involved in HealthWISE activities, such as being relieved from their duties for a short time on a regular basis—perhaps one hour per week—to examine and improve their working environments, while also ensuring that their tasks remained fulfilled and that their departments ran smoothly, might have been one way to overcome this barrier. Taking time to explain the longer-term potential of HealthWISE to the departments might have been another. The resistance by middle management highlighted the importance of communication and consultation across all levels of the hospital and all stakeholders when introducing new interventions to protect HWs. Birken and colleagues [29] developed a theory of the role of middle management in healthcare innovation implementation. Engle and colleagues [30] expanded on this theory and found that middle management in organizations with “high change potential” promoted bidirectional communication and independent thinking and overall supported staff to facilitate implementation. 

Emphasizing the goal of working within limits—of older infrastructure, inadequate human and material resources due to hiring freezes and health budgets, and manual incident reporting—might also have helped to avoid some blockages in the implementation process. Also, given limited data collection and reporting procedures and mechanisms, understanding what type of feedback would have been attainable and useful might also have been helpful, as would have been pointing out synergies with existing organizational priorities and programs and external system regulatory frameworks and legislation. For the most part, workers were enthusiastic and engaged, although some negative attitudes and resistance to change were mentioned. A study by Bergström and colleagues [31] that examined the organizational context construct of the earlier PARiHS framework found that HWs’ commitment to their work, or an “individual’s devotion to the organization,” had an impact on the implementation of knowledge translation interventions in Uganda. Change was less likely where commitment levels were lower and a shortage of human resources contributed to lower levels of commitment. This indicates that where human and possibly other resources are a barrier, organizational culture might need to be addressed and improved for implementation to be successful, particularly in low-resource settings.

There is important interplay between constructs, particularly between “recipients” and “context”. For instance, incentives that contribute to the motivation of participants and project funding for specific equipment and activities might be more effective in some contexts than others. A systematic review by Liu and colleagues [32] found a range of macro, meso, and micro level context factors that influenced the types and impacts of incentives on the recruitment and retention of HWs in multiple countries. In our study, participants from Zimbabwe, where macro level political and socioeconomic factors were straining the country, mentioned that incentives might have been helpful for full participation. They were also initially focused on needing project funding to move forward with any activities and Hospital F was the only facility to indicate that the allocation of funds enabled the implementation of HealthWISE. While our study did not specifically study the effects of externally funded projects, we got the impression that some participants were expecting extra incentives to motivate their participation. We acknowledge, however, that the local conditions and previous experiences with externally funded projects might contribute to this seeming dependency and expectation in LMICs. 

Harvey and colleagues [33] conducted a concept analysis of facilitation and presented the role of the facilitator as one of “supporting people to change their practice”. This HealthWISE study used appointed facilitators, external to the hospitals, who focused on building capacity in OHS in order to enable change, all which fit within the defining characteristics of facilitation that they describe. Hospital A, B, and C in Mozambique considered the HealthWISE trainings useful, but too short, compared with Hospitals D and G in South Africa and Zimbabwe, respectively, which alluded to them only as enabling the implementation of HealthWISE. All ToT workshops were conducted in English; in Mozambique, simultaneous audio translation using headsets and all materials, including the HealthWISE manuals, were available in Portuguese. Additional time may have been warranted due to the language difference, since interpretation demanded a slower pace and longer explanations. For Hospitals A, C, D, E, and F, where participants communicated with the research team, this was perceived as an enabling factor. The process of communication, including tools for translation, ought to have been made clear to participants and questions encouraged, particularly following the ToT workshops to help the initial implementation gain momentum, as opposed to waiting six months for check-in meetings. Overall, the actions carried out by local team members were key to the implementation of HealthWISE. Other studies have demonstrated the critical role of external facilitators and shown links between the different constructs of the PARiHS framework. A study by Ellis and colleagues found that “good facilitation appears to be more influential than context in overcoming the barriers to the uptake of [evidence-based practice]” [34]. This experience helps to show that the roles of the facilitators were key and that improved facilitation across the different constructs could enhance the implementation. For example, facilitators could emphasize both the knowledge and practical aspects of the tool, improving its clarity; they could adapt the training based on workers’ levels of knowledge and skills in regard to OHS; they could demonstrate how the tool fits within the workplace in relation to the hospital’s mission and to other programs and priorities, and help participants advocate for its implementation to improve both worker and patient safety and to better align with existing OHS legislation. 

A key strength of this project was the existing North-South partnership, which enabled the collaboration between multiple countries, one in the Global North and three in the Global South [6]. According to Landau [35], “international research partnerships enact and expose the inequalities, structural constraints, and historically conditioned power relations implicit in the production of knowledge”. Caution is needed and care must be taken to mitigate factors that could exacerbate inequalities and inequities. In this study, local research partners and team members were instrumental in initiating, carrying out, and sustaining the project. One specific measure that was taken was to provide as much direct budget control to the Southern partners as the funder permitted. 

Some limitations were related to administrative issues, such as delays securing approvals from multiple countries and facilities. Data collection instruments ought to also have been piloted, particularly following translation to Portuguese, and streamlined to avoid overburdening participants. This may also have helped to reduce the missing data from incomplete questionnaires; as it stands, further research is required to fully understand the more nuanced aspects of certain constructs, particularly context, where responses were not specific or detailed enough to comprehend how minor differences affected implementation. Understanding how HealthWISE-related changes affected the OHS of HWs was a desired yet difficult-to-capture aspect of this study, particularly due to insufficient existing incident reporting procedures and data collection systems at the participating hospitals. Therefore, while we were able to describe some improvements that were made, we were limited in assessing OHS outcomes and focused instead on enabling factors and barriers to implementation.

As a final reflection on our experience, it is worth noting that the research team initially intended to participate as mere observers to the implementation of HealthWISE, to understand how a standalone tool to improve the health and safety of workers, introduced via a single training session, was implemented in different hospitals in different countries. This hands-off approach was also used to discourage dependency on local teams, whose resources and time are limited, and on international funds, of which there are less and less. However, the capacities of the countries to implement HealthWISE were limited by their differing knowledge and skill levels in regard to OHS; while hospitals in South Africa had workers who specialized in OHS and existing committees, hospitals in Mozambique and even more so in Zimbabwe needed to start with more basic training. Implementation was also limited by resources, which were undeniably scarce in all three countries, but again, especially so in Mozambique and Zimbabwe, as well as by the expectation that external funds were the solution. The team gradually increased their level of facilitation, providing additional practical training and encouraging larger hospitals to start small by focusing on one department instead of being overwhelmed by the whole hospital. This shows that in some contexts, a standalone tool is insufficient and a more robust toolbox that addresses the need to build capacity in implementation science is needed. Future work is focused on developing and testing preparation strategies and materials that might make it easier for individuals and teams who are keen to improve the OHS of their health facilities to do so independently using HealthWISE. Documenting experiences implementing HealthWISE and other tools, including any adaptations made, and making these available via publication or central repository ought to also be encouraged.

## 5. Conclusions

The effort to develop tools for protecting HWs must be accompanied by comparable attention to the way in which these instruments are implemented in settings of need. Our study of the implementation of HealthWISE in seven hospitals in the southern African region provides clear documentation of how a variety of enabling factors and barriers can influence success. Building on methods, such as the application of the i-PARiHS framework which we pursued, a valuable evidence base can be built to support efforts for ensuring that improved work environments for HWs are part of strengthened health systems. With this vision, future implementations ought to focus on areas such as providing clarity about the tool and helping participants to develop clear goals and expectations based on their OHS knowledge and skill levels and on the amount of resources and time available to them. Securing support from senior leadership and middle management and assembling a dedicated and diverse HealthWISE team would also be beneficial. Emphasizing the need to work within existing constraints and find no- or low-cost solutions is also key in resource-poor areas. Future research might focus on examining construct characteristics in further detail and on testing ways to overcome obstacles, as well as on what additional materials might help to create a toolbox, as opposed to a standalone tool, to enable facilities to implement HealthWISE on their own. Promoting and improving the health and safety of HWs at work is part of the solution to increasing recruitment and retention of these essential workers and curbing current and projected shortages in the global health workforce.

## Figures and Tables

**Figure 1 ijerph-17-04519-f001:**
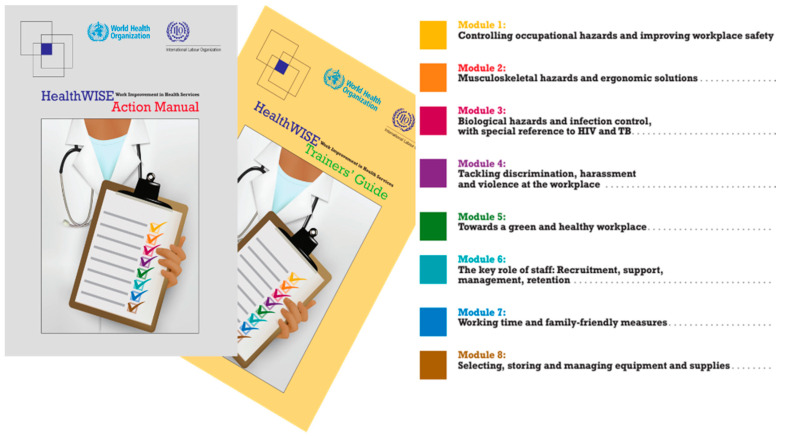
HealthWISE workbooks—the Action Manual and Trainers’ Guide—and a list of the eight modules that cover key areas of occupational health and safety (OHS) for health workers (HWs).

**Figure 2 ijerph-17-04519-f002:**
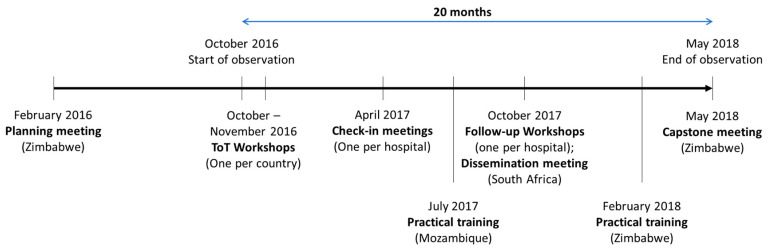
A timeline of the implementation of HealthWISE in seven selected hospitals in Mozambique, South Africa, and Zimbabwe.

**Figure 3 ijerph-17-04519-f003:**
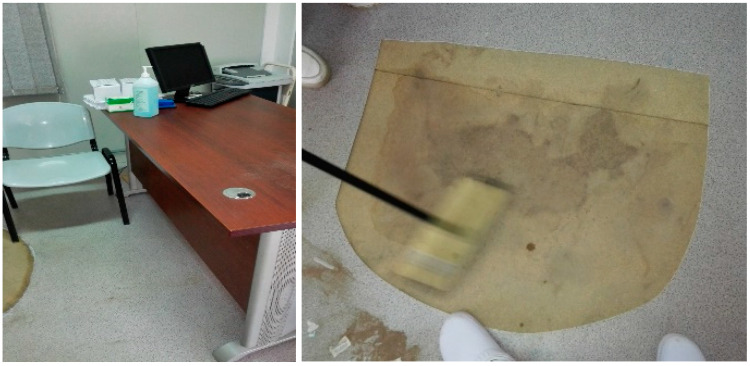
Before (not shown), the physician’s desk was positioned next to the window, with the patient to their right. After, the patient is positioned next to the window and the physician is seated on the opposite side of the desk, allowing the air to flow from the door outside. Ripped flooring has been removed.

**Table 1 ijerph-17-04519-t001:** Characteristics of the seven hospitals (A–G) participating in the HealthWISE project.

Hospital Characteristics	A	B	C	D	E	F	G
Country	MZ	MZ	MZ	SA	SA	ZI	ZI
Number of patient beds	260	272	36	1652	720	1179	160
Number of workers	257	656	139	4407	1930	2843	213
• Male	84	205	32	1146	457	567	78
• Female	173	451	107	3261	1473	2276	135

MZ—Mozambique; SA—South Africa; ZI—Zimbabwe.

**Table 2 ijerph-17-04519-t002:** Innovation construct characteristics and whether they were enabling factors (EF) and/or barriers (B) to the implementation of HealthWISE in each participating hospital (A–G).

Characteristics	A/C	B	D	E	F	G
Clarity	EF	-	-	EF	EF	EF
• Lack of clarity	B	B	-	B	-	-
Relative advantage	EF	-	EF	-	-	-
Observable results	EF	-	-	EF	-	EF

**Table 3 ijerph-17-04519-t003:** Recipient construct characteristics and whether they were enabling factors (EF) and/or barriers (B) to the implementation of HealthWISE in each participating hospital (A–G).

Characteristics	A/C	B	D	E	F	G
Motivation	EF	-	-	-	EF/B	B
Goals and expectations	B	B	EF	EF	-	EF
Skills and knowledge	B	EF	-	-	-	-
Project funding	EF/B	-	B	B	EF/B	B
Human resources	-	B	B	B	B	EF/B
Material resources	B	EF/B	EF	EF	EF	B
Personal protective equipment (PPE)	B	B	-	EF	-	EF
Local opinion leaders	-	-	EF	EF	B	EF
Collaboration and teamwork	EF/B	EF	-	-	-	-
Existing networks	EF	EF	B	EF	-	EF
Power and authority	EF	-	-	B	-	EF

**Table 4 ijerph-17-04519-t004:** Context construct characteristics and whether they were enabling factors (EF) and/or barriers (B) to the implementation of HealthWISE in each participating hospital (A–G).

Characteristics	A/C	B	D	E	F	G
*Local and organizational*						
Senior leadership and management support	EF	EF	EF/B	EF/B	EF/B	EF
• Senior leadership	EF	EF	EF	EF	EF	EF
• Middle management	-	-	B	B	B	-
Culture	B	EF/B	EF/B	B	EF	EF/B
• Commitment to work	-	EF	EF	-	EF	EF
• Knowledge application	-	B	B	B	-	EF
• Worker attitudes/ Resistance to change	B	B	B	B	-	B
Organizational priorities	EF	-	B	-	B	B
Structure and systems	B	B	B	B	B	B
History of innovation and change	B	B	-	B	-	B
Evaluation and feedback processes	EF	B	B	B	-	-
*External health system*						
Regulatory frameworks	-	-	EF	EF	-	-
Environmental (in)stability	-	-	-	-	B	-
Inter-organizational networks/relationships	-	-	EF	EF/B	-	-

**Table 5 ijerph-17-04519-t005:** Facilitation construct characteristics and whether they were enabling factors (EF) and/or barriers (B) to the implementation of HealthWISE in each participating hospital (A–G).

Characteristics	A/C	B	D	E	F	G
HealthWISE trainings	EF/B	B	EF	-	-	EF
Communication with research team	EF	-	EF	EF	EF	-

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
