# Peer review of "Empowering Health Workers to Protect their Own Health: A Study of Enabling Factors and Barriers to Implementing HealthWISE in Mozambique, South Africa, and Zimbabwe"

_ijerph, 2020, doi:10.3390/ijerph17124519_

Round 1

Reviewer 1 Report

Thank you for this very interesting paper relating to the field of implementation science. The approach described by this paper is that of programme fidelity, if I am not mistaken. 

I have a few comments for the authors, which I hope will contribute to improving this very good paper. 

l72: the authors refer to the PARiHS framework, I think it would be relevant for the reader if a short description of this framework appeared at this point. What it this framework? what is its purpose and why is it relevant in this research? 

l77: I would rename the paragraph "country contexts" rather than setting. Also, maybe a short explanation of why the HealthWISE was implemented in the selected countries, and for what purpose, could help the reader follow the red thread. 

general comments:

  • it would be interesting to have more information on the programme. (maybe in the appendixes?) what logic model underlies the programme? what is the programme theory?how is the programme introduced, where does the impulse come from etc.. (as the preceding aspects may also affect further implementation) - if data is available of course
  • Also, what do the authors mean by "implementation "(l99) what was observed exactly? introduction? uptake? implementation of activities? achievement? 

This paper is very clear, interesting and precise. I wongratulate the authors to have submitted such a well-written and relevant paper. the methods used to collect and analyze data are a definite added value that the authors could have discussed and highlighted a little bit more. 

Author Response

Response to Reviewer 1 Comments

Thank you for your considered review of our manuscript.

Point 1: l72: the authors refer to the PARiHS framework, I think it would be relevant for the reader if a short description of this framework appeared at this point. What it this framework? what is its purpose and why is it relevant in this research?

Response 1: While we agree that the framework could be described in further detail here, we have provided a description in lines 154-172 and have added a cross-reference to ‘Research Methods’ in line 81 to indicate that a description is forthcoming. We are attempting to be concise and limit repetition as recommended by another reviewer.

Point 2: l77: I would rename the paragraph "country contexts" rather than setting. Also, maybe a short explanation of why the HealthWISE was implemented in the selected countries, and for what purpose, could help the reader follow the red thread.

Response 2: The paragraph has been renamed ‘Country Contexts’. The following line has been added mid-paragraph in lines 91-92 to emphasise the reasons, alluded to in the ‘Introduction’ for implementing HealthWISE in these three countries: ‘The three countries are in close proximity in the southern African region and represent high-risk settings where the OHS of HWs is at different stages and resource levels and has yet to be fully given the importance that it is due.’

Point 3: it would be interesting to have more information on the programme. (maybe in the appendixes?) what logic model underlies the programme? what is the programme theory?how is the programme introduced, where does the impulse come from etc.. (as the preceding aspects may also affect further implementation) - if data is available of course

Response 3: We have added a sentence referring to the original WISE program developed by the ILO in lines 54-58, on which HealthWISE is based, as well as some details as to the process of its creation (with a reference to the Action Manual from where the information was taken):

‘In 2010, a tripartite group consisting of workers’, employers’ and governments’ representatives, as well as specialists from the ILO and WHO convened and agreed on a framework for improving the OHS of HWs. Based on principles from the original Work Improvement in Small Enterprises (WISE) training program created by the ILO, HealthWISE was then developed to help support the implementation of this framework [4].’

To clarify the logic model associated with the HealthWISE initiative, we have added to the following sentence in lines 58-61: ‘HealthWISE aims to improve working conditions, performance and workplace safety through training and empowering HWs with the ability to identify workplace hazards and areas requiring improvement in their work environments and to conduct processes for developing and implementing low-cost solutions to address them.’

Point 4: Also, what do the authors mean by "implementation "(l99) what was observed exactly? introduction? uptake? implementation of activities? achievement?

Response 4: We have added the following description in lines 110-115 to overview what is meant by ‘implementation’ and ‘observation’ before providing additional details as they were written:

‘Implementation refers to the ensuing activities, including the introduction of HealthWISE by training groups of HWs at participating hospitals and the activities carried out by participants from this point through to the final capstone meeting. Observation focused on if and how participants used HealthWISE in their hospitals and included the activities conducted by research team members, such as focus groups and questionnaires, to inquire into the enabling factors and barriers to its uptake and resultant activities.’

Point 5: This paper is very clear, interesting and precise. I congratulate the authors to have submitted such a well-written and relevant paper. the methods used to collect and analyze data are a definite added value that the authors could have discussed and highlighted a little bit more.

Response 5: Thank you for these supportive comments. We clarified the distinction between implementation of HealthWISE and the methods we used in data collection to facilitate the analysis, including some wording related to the codebook in line 181 as indicated below. Hopefully, it is now clear.

‘Some characteristics were subsequently removed or combined to refine the constructs and characteristics to those listed in Tables 2-5 that were the themes and sub-themes for the final codebook

Reviewer 2 Report

A well written paper finally evaluating the WISE approach. It is a pitty that authors could not provide data on the outcome in terms of improved occupational health of the hospital staff. I recommend that the authors at least discuss this aspect. 

Minor comments:

The discussion should start with a summary of the main findings.

The conclusions are long and repetitive.

Overall, the results and discussion section could be shortened.

Author Response

Response to Reviewer 2 Comments

Thank you for your considered review of our manuscript.

Point 1: A well written paper finally evaluating the WISE approach. It is a pity that authors could not provide data on the outcome in terms of improved occupational health of the hospital staff. I recommend that the authors at least discuss this aspect.

 Response 1: We agree that it is a pity that more information is not available on the effects of HealthWISE-related improvements to health worker OHS. This is due, in part, to the study’s focus on enabling factors and barriers to the implementation of HealthWISE itself (data collection reflects this information, as opposed to OHS outcomes), as well as to insufficient existing incident reporting procedures and data collection systems in the participating hospitals. We have added the following in the ‘Discussion’ section (as part of ‘limitations’) in lines 595-600 to address this:

‘Understanding how HealthWISE-related changes affected the OHS of HWs was a desired yet difficult-to-capture aspect of this study, particularly due to insufficient existing incident reporting procedures and data collection systems at the participating hospitals. Therefore, while we were able to describe some improvements that were made, we were limited in assessing OHS outcomes and focused instead on enabling factors and barriers to implementation.’ 

Point 2: The discussion should start with a summary of the main findings.

Response 2: We have added a summary of the main findings (using similar wording as in the Abstract) to the end of the first paragraph in the ‘Discussion’ in lines 460-468:

‘Applying the i-PARiHS framework enabled an identification of key enabling factors as characteristics of the ‘recipients’ and ‘context’ constructs and included the willingness of workers to engage in implementation, the presence of diverse teams that championed the implementation process, and supportive senior leadership. Barriers were reported in all constructs and included a lack of clarity about how to use HealthWISE, insufficient funds, stretched human resources, older buildings, and lack of incident reporting infrastructure. Overall, successful implementation of HealthWISE called for dedicated local research and technical team members who helped facilitate the process by adapting HealthWISE to the workers’ OHS knowledge and skill levels and the cultures and needs of their hospitals, cutting across all constructs of the i-PARiHS framework.’ 

Point 3: The conclusions are long and repetitive. Overall, the results and discussion section could be shortened.

Response 3: We made every attempt to keep the paper, including with results and discussion sections concise, as we agree that this is important to engage readership from beginning to end. Despite our efforts, the length of the ‘Discussion’ section has admittedly increased based on suggested reviewer revisions, however we hope that this content is desired as opposed to redundant.

Reviewer 3 Report

The authors conducted a qualitative study on the implementation of HealthWISE in three African countries. HealthWISE is a participatory intervention tool aiming to improve occupational health and safety (OSH). Enabling factors and barriers were studied relaying on focus group discussions and other qualitative methods. The study question is relevant for all countries trying to improve the working conditions of health worker (HWs). The methods used are adequate. The results are explained in a concise way and the discussion is relevant.  

The paper can be published in the actual version. However, I propose some minor changes

In the abstract: I propose to introduce the abbreviation i-PARiHS as it is given in line 137 of the manuscript.

Please consider to explain the abbreviation HR in line 295 of the manuscript.

The abbreviation of IPC is explained twice. You might want to delete it in line 366 of the manuscript.

Thank you for the opportunity to read this interesting manuscript.

Author Response

Response to Reviewer 3 Comments

Thank you for your considered review of our manuscript.

Point 1: In the abstract: I propose to introduce the abbreviation i-PARiHS as it is given in line 137 of the manuscript.

Response 1: The full name of the framework ‘integrated Promoting Action on Research Implementation in Health Services’ has been added to the abstract in lines 26-27. It has also been written out in full when first introduced in line 80 and deleted from its later description in line 154.

Point 2: Please consider to explain the abbreviation HR in line 295 of the manuscript.

Response 2: The explanation of ‘[human resources]’ has been added to the text in what is now line 312.

Point 3: The abbreviation of IPC is explained twice. You might want to delete it in line 366 of the manuscript.

Response 3: The second explanation of IPC has been deleted from what is now line 384.